# Expressive Power of Subgraph Graph Neural Networks for Graphs with Bounded Cycles

## Abstract

Graph neural networks (GNNs) have been widely used in graph-related contexts. It is known that the separation power of GNNs is equivalent to that of the Weisfeiler-Lehman (WL) test; hence, GNNs are imperfect at identifying all non-isomorphic graphs, which severely limits their expressive power. This work investigates $k$-hop subgraph GNNs that aggregate information from neighbors with distances up to $k$ and incorporate the subgraph structure. We prove that under appropriate assumptions, the $k$-hop subgraph GNNs can approximate any permutation-invariant/equivariant continuous function over graphs without cycles of length greater than $2k + 1$ within any error tolerance. Our numerical experiments on established benchmarks and novel architectures validate our theory on the relationship between the information aggregation distance and the cycle size.

## 1 Introduction

Graph-based machine learning models, known as graph neural networks (GNNs) (Scarselli et al., 2008; Wu et al., 2020; Zhou et al., 2020; Kipf & Welling, 2016; Veličković et al., 2018), have emerged as powerful tools for interpreting and making predictions on data that can be represented as networks of interconnected points. These models excel at uncovering the underlying structure in graph data, leading to breakthroughs in numerous sectors, including but not limited to physics (Shlomi et al., 2020), chemistry (Reiser et al., 2022; Fung et al., 2021; Coley et al., 2018), bioinformatics (Zhang et al., 2021), finance (Wang et al., 2021), electronic engineering (Liao et al., 2021; He et al., 2021; Lee et al., 2022), and operations research (Gasse et al., 2019).

From a theoretical standpoint, GNNs are utilized to learn or approximate functions on graph-structured data. It is crucial to analyze and understand the expressiveness of GNNs, that is, to determine the class of functions on graphs these networks can effectively approximate. This analysis offers valuable insights that guide the design of more powerful and efficient GNN architectures.

A fundamental concept in the domain of GNNs is the message-passing mechanism (Gilmer et al., 2017), which progressively refines node representations by aggregating information from neighboring nodes. To formalize this process, consider a graph $G$ composed of a vertex set $V = \{v_1, v_2, \ldots, v_n\}$ and an edge set $E$, with each vertex $v_i \in V$ initially endowed with a feature vector $h_i^{(0)}$. The message-passing scheme iteratively updates these vectors. At each iteration or layer, the update for a given vertex integrates input from its immediate neighbors through a combination of two steps: a local transformation and an aggregation. The local transformation applies a learnable function to each neighbor's feature, while the aggregation step combines these transformed features using a permutation-invariant operation such as summing, averaging, or taking the maximum value. This aggregated information is then merged with the current feature of the vertex to produce an updated representation. Mathematically, in the $l$-th iteration/layer, the updated feature $h_i^{(l)}$ for vertex $v_i$ can be expressed as:

$$h_i^{(l)} = f^{(l)}\left(h_i^{(l-1)}, \text{AGGR}\left(\left\{\left\{g^{(l)}(h_j^{(l-1)}) : v_j \text{ adjacent to } v_i\right\}\right\}\right)\right), \tag{1.1}$$

where $f^{(l)}$ and $g^{(l)}$ are learnable functions at layer $l$, and $h_j^{(l-1)}$ represents the feature of vertex $v_j$ from the previous layer. The notation $\{\{\cdot\}\}$ indicates a multiset that can accommodate duplicate elements, ensuring that all contributions from neighboring vertices are considered, even if some vertices share the same feature values.

Despite their empirical successes, message-passing graph neural networks (MP-GNNs) have limitations in terms of separation power or expressive power. Notably, they can fail to differentiate between certain non-isomorphic graphs. For instance, as illustrated in Figure 1, consider two distinct graphs where vertices of the same color start with identical features. Even though these graphs are structurally different, MP-GNNs cannot distinguish between them because vertices of matching colors will end up with the

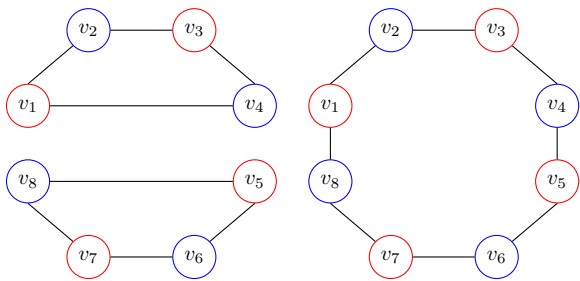

Figure 1: Two non-isomorphic graphs that cannot be distinguished by MP-GNNs or the WL test.

same feature representations after any number of message-passing rounds. This outcome persists irrespective of the specific choices for functions $f^{(l)}, g^{(l)}$, or the aggregation method employed. The reason is that each vertex gathers indistinguishable aggregated information from its neighbors, leading MP-GNNs to perceive these non-isomorphic structures as identical.

The separation and expressive power of MP-GNNs are closely tied to the Weisfeiler-Lehman (WL) test (Weisfeiler & Leman, 1968), a classic algorithm designed to tackle the graph isomorphism problem. At its core, the WL test operates as a color refinement procedure: initially, each vertex $v_i$ is assigned a color $C^{(0)}(v_i)$ based on its initial features. The algorithm then proceeds iteratively by updating the colors according to the following rule:

$$C^{(l)}(v_i) = \text{HASH}\left(C^{(l-1)}(v_i), \left\{\left\{C^{(l-1)}(v_j) : v_j \text{ adjacent to } v_i\right\}\right\}\right), \quad (1.2)$$

which mirrors the structure of the update mechanism in (1.1). Assuming the hash function is collision-free, two vertices will share the same color at iteration $l$ if and only if they have identical colors and multisets of neighbors' colors at iteration $l - 1$. The WL test concludes when the color partition stabilizes, always within no more than $n - 1$ iterations, where $n$ is the number of vertices. It deems two graphs isomorphic if their final color multisets match.

It has been demonstrated that MP-GNNs possess separation power equivalent to that of the Weisfeiler-Lehman (WL) test (Xu et al., 2019). This means that two graphs are identified as non-isomorphic by the WL test if and only if they produce distinct outputs in some MP-GNN. Furthermore, it has been proven in Azizian & Lelarge (2021); Geerts & Reutter (2022) that GNNs can universally approximate any continuous functions whose separation capabilities are bounded above by the associated WL test. However, no polynomial-time algorithms are known to perfectly solve the graph isomorphism problem, implying that the WL test cannot distinguish certain pairs of non-isomorphic graphs, such as those illustrated in Figure 1. As a result, MP-GNNs are unable to represent or approximate all permutation-invariant or permutation-equivariant functions.

In response to these limitations, researchers have proposed alternative GNN architectures designed to enhance separation capabilities. A prominent approach in the literature involves the use of higher-order GNNs (Morris et al., 2019; Maron et al., 2019; Geerts, 2020a;b; Azizian & Lelarge, 2021; Zhao et al., 2022; Geerts & Reutter, 2022; Morris et al., 2020), which correspond to higher-order WL tests (Cai et al., 1992). Essentially, a $k$-th order GNN assigns features to each $k$-tuple of vertices and updates these features based on information from neighboring tuples. This mechanism allows for a more nuanced representation of graph structures, thereby improving the model's ability to distinguish between non-isomorphic graphs.

In this work, we explore an alternative technique that has gained prominence in recent literature (Zhang & Li, 2021; Zhao et al., 2021; Bevilacqua et al., 2021; Frasca et al., 2022) to enhance the expressive power of MP-GNNs. This approach involves incorporating subgraph structures, moving beyond the reliance on vertex features from immediate neighboring nodes alone. Architectures that adopt this strategy are referred to as subgraph GNNs. By integrating subgraph information, these models can capture more complex and nuanced patterns within graph data, thereby improving their ability to distinguish between different graph structures (Feng et al., 2022; Huang et al., 2023). We further perform numerical validation with established benchmarks (Gómez-Bombarelli et al., 2018) and GNN architectures (Ying et al., 2021) in graph learning, whereby a strong correlation between theoretical and numerical results is witnessed.

**Our contributions**   The main contributions of this paper are summarized as follows.

- We rigorously characterize the separation power of subgraph GNNs by demonstrating that they can perfectly distinguish a large family of graphs with bounded cycle lengths. Specifically, under appropriate assumptions, we prove that GNNs leveraging subgraph structures within a distance $k$ from the current vertex can distinguish all non-isomorphic graphs that do not contain cycles of length greater than $2k + 1$.
- Based on the above result, we show that such $k$-hop subgraph GNNs can approximate any permutation-invariant/equivariant continuous functions on graphs without cycles of length greater than $2k + 1$.
- We empirically validate our theoretical findings on the relationship between the information aggregation distance $k$ and the cycle size $2k + 1$, providing valuable insights on selecting practical information aggregation distance.

**Organization**   The rest of this paper will be organized as follows. We define subgraph GNNs and the associated WL test and introduce the motivation in Section 2. Our main theory for the expressive power of $k$-hop subgraph GNNs is presented in Section 3, with detailed proofs and extensions deferred to the appendix. Section 4 presents the numerical experiments and the whole paper is concluded in Section 5.

## 2 SUBGRAPH GNNS AND THE WL TEST

This section describes the motivation and sets up the basics for subgraph GNNs.

### 2.1 MOTIVATION

Note that MP-GNNs (1.1) have a fundamental limitation that they fail to distinguish some non-isomorphic graphs, such as those in Figure 1. One idea to enhance the expressive power is to incorporate more information from neighboring vertices and subgraphs:

- The aggregation in (1.1) uses $\mathcal{N}(v_i)$, the set of neighbors of $v_i$. To incorporate additional information, we define $d(u, v)$ as the shortest-path distance between $u$ and $v$ in the graph $G$ and
$$\mathcal{N}_k(v_i) := \{v \in G : d(v, v_i) \leq k\}, \quad k \geq 1.$$
- Beyond the vertex features in $\mathcal{N}_k(v_i)$, one can also capture edge information, i.e., whether two vertices are connected. This means that the topology of $G|_{\mathcal{N}_k(v_i)}$, the subgraph of $G$ restricted to $\mathcal{N}_k(v_i)$ (known as the $k$-hop subgraph rooted at $v_i$), can be used to update the feature of $v_i$.

Let $(G, h^{(l-1)})_{v_i,k}$ denote the subgraph $G|_{\mathcal{N}_k(v_i)}$ rooted at $v_i$, with each vertex having a feature from $h^{(l-1)}$. Accordingly, the vertex feature update rule is given by

$$h_i^{(l)} = f^{(l)} \left( h_i^{(l-1)}, g^{(l)} \left( (G, h^{(l-1)})_{v_i,k} \right) \right). \tag{2.1}$$

The functions $f^{(l)}$ and $g^{(l)}$ are learnable, with $g^{(l)}$ taking the same value on isomorphic rooted graphs. This scheme, termed the *$k$-hop subgraph GNN*, has various applications and adaptations in the existing literature (Zhang & Li, 2021; Zhao et al., 2021; Bevilacqua et al., 2021; Frasca et al., 2022). Notably, the learnable function $g^{(l)}$ is often parameterized as another GNN applied to the smaller subgraph $(G, h^{l-1})_{v_i,k}$.

Consider the two non-isomorphic graphs in Figure 1 indistinguishable by MP-GNNs. It can be seen that the 2-hop subgraph GNN can successfully distinguish them. Specifically, the 2-hop subgraphs rooted at $v_1$ are shown in Figure 2 and are clearly non-isomorphic, indicating that $v_1$ in the two graphs in Figure 1 will have different feature after one layer of the 2-hop subgraph GNN, as long as $g^{(1)}$ can distinguish these two non-isomorphic subgraphs.

Another observation is that the 2-hop subgraph GNN fails if we increase the cycle sizes in Figure 1—for instance, by changing one graph to have two 6-cycles and the other a single 12-cycle. In general, the $k$-hop subgraph GNN fails to distinguish between a graph with two $(2k + 2)$-cycles and one with a single $(4k + 4)$-cycle, though it succeeds when the cycle sizes are smaller. This suggests that larger cycles limit the separation power of the $k$-hop subgraph GNN.

Such observation and motivation align with empirical findings in the literature. The ZINC dataset (Dwivedi et al., 2023) consists of molecular graphs with no large cycles, and variants of subgraph GNNs have shown notable improvement over message-passing GNNs on this dataset (Zhang et al., 2023; Zhang & Li, 2021; Zhao et al., 2021; Bevilacqua et al., 2021; Frasca et al., 2022).

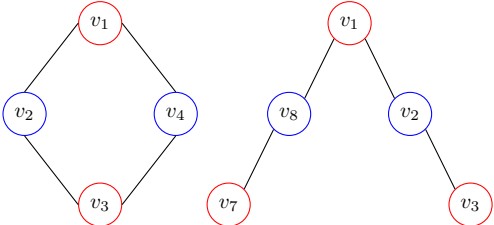

Figure 2: 2-hop subgraphs rooted at $v_1$ for graphs in Figure 1

At the end of this subsection, we comment that the above observation on $k$-hop subgraph GNNs requires that $g^{(l)}$ has a relatively strong separation power on the subgraph $(G, h^{(l-1)})_{v_i,k}$, which is achievable if the size of $(G, h^{(l-1)})_{v_i,k}$ is small or $G$ is sparse. However, if $G$ is relatively dense, then the size of $(G, h^{(l-1)})_{v_i,k}$ might be comparable with $G$ and the subproblem of finding expressive $g^{(l)}$ might be difficult.

### 2.2 $k$-HOP SUBGRAPH GNNs

We rigorously define the $k$-hop subgraph GNNs in this subsection, for which we define the graph space first.

**Definition 2.1** (Space of graphs with vertex features). *We use $\mathcal{G}_{n,m}$ to denote the space of all undirected unweighted graphs of $n$ vertices with each vertex equipped with a feature in $\mathbb{R}^m$. The space $\mathcal{G}_{n,m}$ is equipped with the product topology of discrete topology (of graphs without vertex features) and Euclidean topology (of vertex features).*

We use $(G, H)$ to denote an element in $\mathcal{G}_{n,m}$ where $G = (E, V)$ is an undirected unweighted graph, and $H = (h_1, h_2, \ldots, h_n)$ is the collection of all vertex features. Given $(G, H) \in \mathcal{G}_{n,m}$ and $k \geq 1$, the $k$-hop subgraph GNN is defined as follows.

- The embedding layer maps each vertex feature $h_i \in \mathbb{R}^m$ as an embedding vector
$$h_i^{(0)} = f^{(0)}(h_i),$$
where $f^{(0)}$ is learnable.

- For $l = 1, 2, \ldots, L$, the information aggregation layer computes $h_i^{(l)}$ according to (2.1) for $i = 1, 2, \ldots, n$.

- There are two types of outputs. The graph-level output computes a real number for the whole graph, namely
$$y = r\left(\mathrm{AGGR}\left(\left\{\left\{h_i^{(L)} : i \in \{1, 2, \ldots, n\}\right\}\right\}\right)\right),$$
where $r$ is learnable. The vertex-level output assigns a real number for each vertex:
$$y_i = r(h_i^{(L)}), \quad i = 1, 2, \ldots, n.$$

In general, the intermediate vertex features $h_i^{(l)}$ can be defined in any topological space, while one usually uses Euclidean spaces in practice. Throughout this paper, we always consider continuous $f^{(l)}$, $g^{(l)}$, and $r$, guaranteeing that all $k$-hop subgraph GNNs are continuous.

**Definition 2.2** (Spaces of $k$-hop subgraph GNNs). *We use $\mathcal{F}_k$ to denote the collection of all $k$-hop subgraph GNNs with graph-level output, and use $\mathcal{F}_{k,v}$ to denote the collection of all $k$-hop subgraph GNNs with vertex-level output.*

It is clear that a $k$-hop subgraph GNN with graph-level output is permutation-invariant, and a $k$-hop subgraph GNN with vertex-level output is permutation-equivariant, with respect to the following definition.

**Definition 2.3** (Permutation-invariant and permutation-equivariant functions). *We say that a function $\Phi : \mathcal{G}_{n,m} \to \mathbb{R}$ is permutation-invariant if*
$$\Phi(\sigma * (G, H)) = \Phi(G, H), \quad \forall \sigma \in S_n,$$

*where $S_n$ denotes the permutation group on $\{1, 2, \ldots, n\}$ and $\sigma * (G, H)$ is the graph obtained by relabeling vertices in $(G, H)$ according to the permutation $\sigma$, and that a function $\Phi : \mathcal{G}_{n,m} \to \mathbb{R}^n$ is permutation-equivariant if*

$$\Phi(\sigma * (G, H)) = \sigma(\Phi(G, H)), \quad \forall \sigma \in S_n.$$

## 2.3 $k$-HOP SUBGRAPH WL TEST AND THE EQUIVALENT SEPARATION POWER

It is known that the MP-GNNs and the classic WL test have equivalent separation power (Xu et al., 2019). The same equivalence extends naturally for other variants of GNNs and WL test. In particular, we consider the $k$-hop subgraph GNN and the associated WL test stated in Algorithm 1.

We define two equivalence relationships below that characterize the separation power of the $k$-hop subgraph WL test.

**Definition 2.4.** *For two graphs $(G, H), (\hat{G}, \hat{H}) \in \mathcal{G}_{n,m}$, denote $\{\{C_i^{(L)} : i \in \{1, 2, \ldots, n\}\}\}$ and $\{\{\hat{C}_i^{(L)} : i \in \{1, 2, \ldots, n\}\}\}$ as their final color multisets output by the $k$-hop subgraph WL test.*

---

**Algorithm 1** $k$-hop Subgraph Weisfeiler-Lehman test

**Require:** A graph $(G, H) \in \mathcal{G}_{n,m}$ and iteration limit.
Initialize the vertex color

$$C_i^{(0)} = \text{HASH}(h_i), \quad i = 1, 2, \ldots, n.$$

**while** $l = 1, 2, \ldots, L$ **do**
  Refine the color

$$C_i^{(l)} = \text{HASH}\left(C_i^{(l-1)}, (G, C^{(l-1)})_{v_i, k}\right), \quad (2.2)$$

  for $i = 1, 2, \ldots, n$.
**end while**
**Output:** Multiset $\{\{C_i^{(L)} : i \in \{1, 2, \ldots, n\}\}\}$.

---

   *(i) We say $(G, H) \overset{k}{\sim} (\hat{G}, \hat{H})$ if $\{\{C_i^{(L)} : i \in \{1, 2, \ldots, n\}\}\} = \{\{\hat{C}_i^{(L)} : i \in \{1, 2, \ldots, n\}\}\}$ for any $L > 0$ and any hash function.*

   *(ii) We say $(G, H) \overset{k,v}{\sim} (\hat{G}, \hat{H})$ if $C_i^{(L)} = \hat{C}_i^{(L)}$, $i = 1, 2, \ldots, n$, for any $L > 0$ and any hash function.*

We remark that two multisets are identical if for any element, its multiplicities in two multisets are the same. Throughout this paper, we would say that two graphs $(G, H)$ and $(\hat{G}, \hat{H})$ are indistinguishable by $k$-hop subgraph WL test if $(G, H) \overset{k}{\sim} (\hat{G}, \hat{H})$.

The following theorem states the equivalence between the separation power of $k$-hop subgraph GNNs and the $k$-hop subgraph WL test.

**Theorem 2.5.** *For any $(G, H), (\hat{G}, \hat{H}) \in \mathcal{G}_{n,m}$ and any $k > 0$, the following are equivalent:*

   *(i) $(G, H) \overset{k}{\sim} (\hat{G}, \hat{H})$.*

   *(ii) $F(G, H) = F(\hat{G}, \hat{H})$ for any $F \in \mathcal{F}_k$.*

   *(iii) For any $F_v \in \mathcal{F}_{k,v}$, there exists $\sigma \in S_n$ such that $F_v(G, H) = \sigma(F_v(\hat{G}, \hat{H}))$.*

*Moreover, $(G, H) \overset{k,v}{\sim} (\hat{G}, \hat{H})$ if and only if $F_v(G, H) = F_v(\hat{G}, \hat{H})$ for any $F_v \in \mathcal{F}_{k,v}$.*

The proof of Theorem 2.5 follows similar lines as in the proof of Theorem 4.2 in Chen et al. (2023), which is straightforward and is hence omitted. Similar equivalence also holds for two vertices in a single graph, as stated in the corollary below.

**Corollary 2.6.** *For any $(G, H) \in \mathcal{G}_{n,m}$ and any $k > 0$. Let $\{\{C_i^{(L)} : i \in \{1, 2, \ldots, n\}\}\}$ be the color multiset output by the $k$-hop subgraph WL test. For any $i, i' \in \{1, 2, \ldots, n\}$, the following are equivalent:*

   *(i) $C_i^{(L)} = C_{i'}^{(L)}$ for any $L > 0$ and any hash function.*

   *(ii) $F_v(G, H)_i = F_v(G, H)_{i'}$ for any $F_v \in \mathcal{F}_{k,v}$.*

*Proof.* Apply Theorem 2.5 to $(G, H)$ and $\sigma * (G, H)$ where $\sigma$ is the permutation that switches $i, i'$ and keep all other indices unchanged. $\qquad \square$

## 3 EXPRESSIVE POWER OF $k$-HOP SUBGRAPH GNNS

This section presents our main results on the expressive power of $k$-hop subgraph GNNs. We consider $k = 1$ in Section 3.1 and $k \geq 2$ in Section 3.2.

### 3.1 1-HOP SUBGRAPH GNNS

Our main theorem on 1-hop subgraph GNNs is that they can approximate any permutation-invariant/equivariant continuous functions on graphs without cycles of length greater than 3.

**Theorem 3.1.** *Let $\mathbb{P}$ be a Borel probability measure on $\mathcal{G}_{n,m}$. Suppose that for $\mathbb{P}$-almost surely $(G, H)$, the graph $G$ is connected and has no cycles of length greater than 3. Then, the following hold.*

  *(i) For any $\epsilon, \delta > 0$ and any permutation-invariant continuous function $\Phi : \mathcal{G}_{n,m} \to \mathbb{R}$, there exists $F \in \mathcal{F}_1$ such that*

$$\mathbb{P}\left[ |F(G, H) - \Phi(G, H)| > \delta \right] < \epsilon.$$

  *(ii) For any $\epsilon, \delta > 0$ and any permutation-equivariant continuous function $\Phi_v : \mathcal{G}_{n,m} \to \mathbb{R}^n$, there exists $F_v \in \mathcal{F}_{1,v}$ such that*

$$\mathbb{P}\left[ \|F_v(G, H) - \Phi_v(G, H)\| > \delta \right] < \epsilon.$$

Throughout this paper, we always denote $\| \cdot \|$ as the standard $\ell_2$-norm on $\mathbb{R}^n$. We describe the main idea here and the detailed proof of Theorem 3.1 is deferred to Section A. The classic Stone-Weierstrass theorem states that under mild conditions, a function class can universally approximate any continuous function if and only if it separates points, i.e., for any two different inputs, at least one function in that class has different outputs. Therefore, based on Stone-Weierstrass-type theorems, it suffices to show that 1-hop subgraph GNNs have strong enough separation power to distinguish all non-isomorphic connected graphs with no cycles of length greater than 3. Noticing the equivalence results in Section 2.3, one only needs to explore the separation power of 1-hop subgraph WL test.

**Theorem 3.2.** *Consider $(G, H), (\hat{G}, \hat{H}) \in \mathcal{G}_{n,m}$. Suppose that $G$ and $\hat{G}$ are both connected and have no cycles of length greater than 3. If $(G, H) \overset{1}{\sim} (\hat{G}, \hat{H})$, then $(G, H)$ and $(\hat{G}, \hat{H})$ must be isomorphic.*

In the acyclic graph setting, it is proved in Bamberger (2022) that two trees indistinguishable by the classic WL test (1.2) must be isomorphic. Theorem 3.2 can be viewed as a generalization of this result from Bamberger (2022).

The key idea in the proof of Theorem 3.2 is inductively constructing the isomorphism. We consider stabilized colors output by 1-hop subgraph WL test without hash collision and start from two vertices of the same color, one from each graph. The 1-hop subgraphs rooted at these two vertices are isomorphic, guaranteed by the same color. Then we inductively extend the subgraphs by adding neighbors of two vertices in the current subgraphs of the same color, which maintains the isomorphism, until they reach the whole graphs.

### 3.2 $k$-HOP SUBGRAPH GNNS WITH $k \geq 2$

This subsection concerns the expressive power of $k$-hop subgraph GNNs for $k \geq 2$ and the main theory is an extension of Theorem 3.1, in the sense that $k$-hop subgraph GNNs ($k \geq 2$) can approximate any permutation-invariant/equivariant continuous functions on graphs without cycles of length greater than $2k + 1$, but an additional assumption is required.

**Definition 3.3** ($k$-separable graph). *A graph $(G, H) \in \mathcal{G}_{n,m}$ is said to be $k$-separable if the following condition holds when the $k$-hop subgraph WL test terminates with stabilized colors and without hash collisions: For any three vertices $u, v_1, v_2$ with $d(u, v_1) = d(u, v_2) = k$ and $v_1 \neq v_2$, the colors of $v_1$ and $v_2$ are different.*

**Theorem 3.4.** *Let $\mathbb{P}$ be a Borel probability measure on $\mathcal{G}_{n,m}$. Suppose that $\mathbb{P}$-almost surely, $(G, H)$ is $k$-separable and $G$ is connected with no cycles of length greater than $2k + 1$. Then, the following hold.*

(i) *For any $\epsilon, \delta > 0$ and any permutation-invariant continuous function $\Phi : \mathcal{G}_{n,m} \to \mathbb{R}$, there exists $F \in \mathcal{F}_k$ such that*

$$\mathbb{P}\left[|F(G, H) - \Phi(G, H)| > \delta\right] < \epsilon.$$

(ii) *For any $\epsilon, \delta > 0$ and any permutation-equivariant continuous function $\Phi_v : \mathcal{G}_{n,m} \to \mathbb{R}^n$, there exists $F_v \in \mathcal{F}_{k,v}$ such that*

$$\mathbb{P}\left[\|F_v(G, H) - \Phi_v(G, H)\| > \delta\right] < \epsilon.$$

The proof of Theorem 3.4 follows a similar framework as Theorem 3.1 and is deferred to Section B, where the key step is the following theorem that is an analog of Theorem 3.2.

**Theorem 3.5.** *Consider $k \geq 2$ and $(G, H), (\hat{G}, \hat{H}) \in \mathcal{G}_{n,m}$ that are both $k$-separable. Suppose that $G$ and $\hat{G}$ are both connected and have no cycles of length greater than $2k + 1$. If $(G, H) \overset{k}{\sim} (\hat{G}, \hat{H})$, then $(G, H)$ and $(\hat{G}, \hat{H})$ must be isomorphic.*

We remark that even restricted to $k$-separable graphs, the $k$-hop subgraph WL test (2.2) still has strictly stronger separation power compared to the classic WL test (1.2). To illustrate this, we present two non-isomorphic $k$-separable graphs that can be distinguished by the $k$-hop subgraph WL test, but are, however, treated the same by the classic WL test. Let $k = 3$, and consider the two graphs in Figure 3 with initial vertex features as labeled by colors.

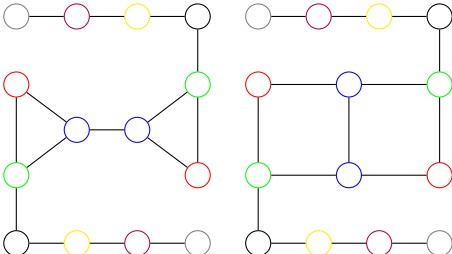

Figure 3: Two non-isomorphic 3-separable graphs indistinguishable by the classic WL test, but distinguishable by the 3-hop subgraph WL test.

Notice that neither graph has a cycle with more than $2k + 1 = 7$ vertices. Furthermore, for any vertex $u$ in either graph, any distinct vertices $v_1$ and $v_2$ with distance exactly 3 from $u$ are of different colors. Thus, our results imply that these two graphs can be distinguished by the 3-hop subgraph WL test. However, we can see that the coloring on both graphs immediately stabilizes when the classic WL test is applied, so the classic WL test cannot distinguish between the graphs. Moreover, this example is non-trivial in the sense that any 3-hop subgraph in either graph is not the entire graph.

At the end of this section, we will mention some related works that analyze the separation power of the subgraph GNNs. Feng et al. (2022) show that the separation power of subgraph GNNs is partially stronger than the third-order WL test. More related to our work, it is proved in Huang et al. (2023) that subgraph GNNs can "count" cycles of length up to 4 and some variant can "count" cycles of length up to 6. It is worth noting that the subgraph topology is integrated in a specific way in Feng et al. (2022); Huang et al. (2023), while we always assume that $g^{(l)}$ has strong enough expressive/separation power on the $k$-hop subgraphs, without fixing the structure of $g^{(l)}$.

## 4 NUMERICAL EXPERIMENTS

### 4.1 EXPERIMENTAL SETTING AND DATASET

Our experiments utilize a special family of $k$-hop subgraph GNNs that implements (2.1) in a practical and computationally cheap way, namely

$$h_i^{(l)} = f^{(l)}\left(h_i^{(l-1)}, \text{AGGR}\left(\left\{\left\{g^{(l)}\left(h_j^{(l-1)}, d(v_i, v_j)\right) : v_j \in \mathcal{N}_k(v_i)\right\}\right\}\right)\right), \qquad (4.1)$$

where the vertex feature is updated by aggregating information in $\mathcal{N}_k(v_i)$ and some information of the subgraph topology is integrated in a shortest-path distance $d(v_i, v_j)$ on the graph. In particular, we implement a family of models termed $k$-hop Graphormer and modified from the official Graphormer backbone (Ying et al., 2021). We also show the generalizability of our framework by implementing the $k$-hop Graphormer-GD with a different definition of $d(v_i, v_j)$ (Zhang et al., 2023). The numerical experiment aims to validate our theoretical findings on an established graph learning

benchmark using state-of-the-art GNN backbones. We also aim to assess whether the theoretically characterized expressiveness could generalize from the training set to the testing set, which is not characterized by most GNN-related theories, nor the recent finding that many graph benchmarks are empirically distinguishable by either 1-WL (Zopf, 2022) or subgraph counting (Pellizzoni et al., 2025). All experiments are performed on a server with Intel 6230R CPU, single 2080Ti GPU, and 512GB RAM.

We test our model on the ZINC graph-learning dataset, a subset of the ZINC database with 250,000 molecules developed by Gómez-Bombarelli et al. (2018) (often denoted as ZINC-Full). On this dataset, the aim of the machine learning community mainly focuses on predicting the water-octanol partition coefficient (i.e., logP (Wildman & Crippen, 1999)) by neural networks, whereby the ground truth labels are computed with cheminformatics tools. As shown in Figure 4, most graphs in the ZINC testing dataset have a longest cycle no larger than 10. Given these statistics and considering our theoretical bounds on GNN expressiveness for cycle sizes no more than $2k + 1$, we anticipate a notable performance boost around $k = 4$ or $k = 5$ for our $k$-hop Graphormers. In line with peer methods, we report the mean absolute error (MAE) between GNN predictions and ground-truth labels.

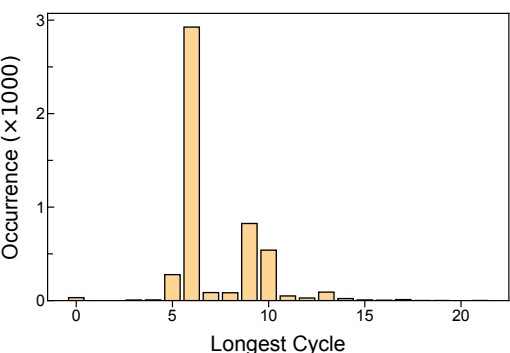

Figure 4: Statistics on the longest cycle length across all testing molecules in the ZINC dataset indicate that most molecules have a longest cycle of 6, aligning with the chemical intuition that 6-membered rings are particularly stable. The peaks observed at 9 and 10 also support the prevalence of common fused ring systems.

### 4.2 $k$-HOP GRAPHORMER IMPLEMENTATION DETAILS

Graphormer replaces edge-wise message-passing with attention layers, whereby the connectivity information is incorporated into attention weights, which are initialized with pairwise shortest path distance. We modify its attention mechanism to fit Graphormers with our $k$-hop theory, by masking out all attention paths that are beyond the $k$-hop neighbors of the aggregated node. This implementation is in line with the aggregation function in (4.1) and $d(v_i, v_j)$ is the shortest path distance. Graphormer-GD further improves expressiveness by replacing the shortest path distance with resistance distance (Klein & Randic, 1993), which could also be incorporated with our $k$-hop architecture. Graphormer(-GD) is a special case of $k$-hop Graphormer(-GD) by setting $k = \infty$. We reimplement and test all Graphormers with the following default hyperparameters: Graphormer-Slim backbone, Adam with $2 \times 10^{-4}$ starting learning rate and

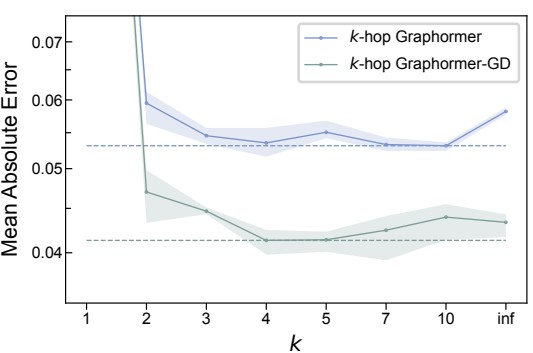

Figure 5: Mean absolute error (MAE) on ZINC testing set with different $k$s for our $k$-hop Graphormer(-GD). Dashed horizontal lines denote the best-performing entry. Aligned with theoretical results and dataset statistics in Figure 4, we see a significant performance boost from $k = 1$ to 4, and the performance is nearly saturated for $k \geq 4$. All results are obtained with 3 random seeds, and the min/max MAEs are shown.

ending $10^{-9}$, 12 Transformer layers, 80 hidden dimensions, 8 attention heads, 60,000 warm-up steps, and 400,000 total steps. To mitigate the gap between the slight implementation difference of the computationally-friendly (4.1) between (2.1), we derive similar theoretical results for cycle lengths of $2k - 1$ in Section C.

### 4.3 RESULTS AND DISCUSSIONS

In the experiments, we evaluate the $k$-hop Graphormers(-GD) with $k = 1, 2, 3, 4, 5, 7, 10$, and infinity, i.e., feature could be aggregated from all nodes by attention. As shown in Figure 5, 1-hop Graphormer performs relatively inferior, and the performance improves significantly with larger $k$ when $k \leq 4$ and reach a nearly optimal performance at $k = 4$, validating our theoretical discovery that a $k$-hop Graphormer has the expressiveness to

Table 1: Mean Absolute Error (MAE) on ZINC test set. The best entry among Graphormer variants is underlined, and the best entry among Graphormer-GD variants is highlighted in **bold**.

| Aggregation | Model | Test MAE ↓ |
|---|---|---|
| 1-hop | GIN (Xu et al., 2019) | 0.088±0.002 |
| | GraphSAGE (Hamilton et al., 2017) | 0.126±0.003 |
| | GAT (Veličković et al., 2018) | 0.111±0.002 |
| | $k$-**hop Graphormer** ($k = 1$) | 0.459±0.001 |
| $k$-hop | $k$-**hop Graphormer** ($k = 4$) | 0.054±0.002 |
| | $k$-**hop Graphormer** ($k = 7$) | 0.053±0.001 |
| | $k$-**hop Graphormer-GD** ($k = 4$) | **0.041±0.001** |
| | $k$-**hop Graphormer-GD** ($k = 7$) | 0.042±0.001 |
| full-graph | Graphormer (Ying et al., 2021) | 0.058±0.001 |
| | Graphormer-GD (Zhang et al., 2023) | 0.043±0.001 |

learn graphs with longest cycles of $2k + 1$. The performance becomes nearly saturated between $k = 4$ and 10, which is within expectation as the marginal improvements in covered graphs are less significant if read together with Figure 4. $k$-hop Graphormer-GD benefits from the resistance distance over the shortest path distance, validating the effectiveness of $k$-hop design on both distance metrics. We also see a slight performance drop for $k$-hop Graphormer in the special case of $k = \infty$, indicating that too many message paths may introduce more noise than information to the model performance. A smaller $k$ also brings potential efficiency improvements as it has fewer message-passing paths than an infinite $k$. The efficiency benefit is not obvious for the ZINC dataset but will become valuable for larger-sized graphs.

When compared with other 1-hop aggregation methods (traditional message-passing GNNs), as shown in Table 1, our 1-hop Graphormer has a similar MAE in scale but is inferior to 1-hop GNNs. With an appropriate $k$ that reflects the distribution of cycle lengths in data, $k$-hop Graphormers reach a significant improvement. Our model degenerates to the original Graphormer when $k \to \infty$. It is also worth noting that this benchmark is nearly saturated, for example, Zhang et al. (2023) reported a 0.025 MAE with a thorough search of the hyperparameter space but their official implementation does not include the hyperparameters to reproduce it. When training Graphormer-GD with the same hyperparameters from the original Graphormer, the average MAE becomes 0.043. Considering all of these, we believe it is beyond the scope of this paper to develop new state-of-the-art. Therefore, our main focus of experiments is to validate our theoretical results, i.e., the correlation between $k$ values and the performance of $k$-hop Graphormers.

## 5 CONCLUSION

This paper rigorously evaluates the efficiency of GNNs that leverage subgraph structures, particularly on graphs with bounded cycles, which represent many real-world datasets. In particular, we prove that $k$-hop subgraph GNNs can reliably predict properties of graphs without cycles of length greater than $2k + 1$, which is unconditionally if $k = 1$ and requires an additional assumption for $k \geq 2$. The correlation between $k$-hop subgraph GNNs and $2k + 1$ cycle size is further validated by numerical experiments.

Let us also comment on the limitations of the current work. Firstly, it is unclear whether the $k$-separability assumption in Theorem 3.4 can be removed or not. Secondly, Theorem 3.1 and Theorem 3.4 rely on the strong enough separation power of $g^{(l)}$ in (2.1) on the $k$-hop subgraphs, which might be expensive to guarantee if the input graph is dense, and it is still open whether one can establish theories of a similar style for specific information aggregation mechanisms on $k$-hop subgraphs, such as (4.1) and Feng et al. (2022); Huang et al. (2023). We have included some preliminary results for the expressive power of (4.1) in Appendix C. Lastly, it is also interesting to test $k$-hop subgraph GNNs on more real-world datasets. These directions deserve future research.

ACKNOWLEDGMENTS

We used large language models (ChatGPT and DeepSeek) as tools for minor language editing. All technical content was written and verified by the authors.

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

## A    PROOFS FOR SECTION 3.1

The proof of Theorem 3.1 is based on Theorem 3.2 and its corollary.

**Corollary A.1.** *Consider any* $(G, H) \in \mathcal{G}_{n,m}$ *where* $G$ *is connected and has no cycles of length greater than 3. Let* $\{\{C_i^{(L)} : i \in \{1, 2, \ldots, n\}\}\}$ *be the color multiset output by the* 1-hop *subgraph WL test. For any* $i, i' \in \{1, 2, \ldots, n\}$, *if* $C_i^{(L)} = C_{i'}^{(L)}$ *holds for any* $L > 0$ *and any hash function, then we have for any permutation-equivariant function* $\Phi : \mathcal{G}_{n,m} \to \mathbb{R}^n$ *that* $\Phi(G, H)_i = \Phi(G, H)_{i'}$.

We will postpone the proofs of Theorem 3.2 and Theorem A.1 and first prove Theorem 3.1 (i) using Theorem 3.2 and the Stone-Weierstrass theorem.

*Proof of Theorem 3.1 (i).* There exists a compact and permutation-invariant subset $X \subseteq \mathcal{G}_{n,m}$ such that $\mathbb{P}[X] > 1 - \epsilon$ and that for any $(G, H) \in X$, $G$ is connected and has no cycles of length greater than 3. Due to Theorem 3.2 and the permutation-invariant property of $\Phi$, $\Phi|_X : X \to \mathbb{R}$ induces a continuous map on the quotient space $\widetilde{\Phi|_X} : X/ \overset{1}{\sim} \to \mathbb{R}$ By the same reason, for $F \in \mathcal{F}_1$, $F|_X : X \to \mathbb{R}$ also induces a continuous map $\widetilde{F|_X} : X/ \overset{1}{\sim} \to \mathbb{R}$. Consider any $(G, H), (\hat{G}, \hat{H}) \in X$ that represent different elements in $X/ \overset{1}{\sim}$, Theorem 2.5 guarantees that there exists $F \in \mathcal{F}_1$ such

that $F(G, H) \neq F(\hat{G}, \hat{H})$, suggesting that $\{\widetilde{F|_X} : F \in \mathcal{F}_1\}$ separates points on $X/\overset{1}{\sim}$. Therefore, by the Stone-Weierstrass theorem, one can conclude that there exists $F \in \mathcal{F}_1$ such that

$$\left\| \widetilde{F|_X} - \widetilde{\Phi|_X} \right\|_{L^\infty(X/\overset{1}{\sim})} < \delta,$$

which implies that

$$|F(G, H) - \Phi(G, H)| < \delta, \quad \forall (G, H) \in X.$$

Thus, it holds that

$$\mathbb{P}\left[|F(G, H) - \Phi(G, H)| > \delta\right] \leq \mathbb{P}[\mathcal{G}_{n,m} \backslash X] < \epsilon,$$

which completes the proof. $\qquad\square$

The proof of Theorem 3.1 (ii) requires a generalized Stone-Weierstrass theorem for equivariant functions.

**Theorem A.2** (Generalized Stone-Weierstrass theorem, Theorem 22 in Azizian & Lelarge (2021)). *Let $X$ be a compact topological space and let $\mathbf{G}$ be a finite group that acts continuously on $X$ and $\mathbb{R}^n$. Define the collection of all equivariant continuous functions from $X$ to $\mathbb{R}^n$ as follows:*

$$\mathcal{C}_e(X, \mathbb{R}^n) = \{F \in \mathcal{C}(X, \mathbb{R}^n) : F(g * x) = g * F(x), \, \forall \, x \in X, g \in \mathbf{G}\}.$$

*Consider any $\mathcal{F} \subset \mathcal{C}_e(X, \mathbb{R}^n)$ and any $\Phi \in \mathcal{C}_e(X, \mathbb{R}^n)$. Suppose the following conditions hold:*

*(i) $\mathcal{F}$ is a subalgebra of $\mathcal{C}(X, \mathbb{R}^n)$ and $\mathbf{1} \in \mathcal{F}$.*

*(ii) For any $x, x' \in X$, if $f(x) = f(x')$ holds for any $f \in \mathcal{C}(X, \mathbb{R})$ with $f\mathbf{1} \in \mathcal{F}$, then for any $F \in \mathcal{F}$, there exists $g \in \mathbf{G}$ such that $F(x) = g * F(x')$.*

*(iii) For any $x, x' \in X$, if $F(x) = F(x')$ holds for any $F \in \mathcal{F}$, then $\Phi(x) = \Phi(x')$.*

*(iv) For any $x \in X$, it holds that $\Phi(x)_i = \Phi(x)_{i'}, \, \forall \, (i, i') \in I(x)$, where*

$$I(x) = \left\{(i, i') \in \{1, 2, \dots, n\}^2 : F(x)_i = F(x)_{i'}, \, \forall \, F \in \mathcal{F}\right\}.$$

*Then for any $\epsilon > 0$, there exists $F \in \mathcal{F}$ such that*

$$\sup_{x \in X} \|\Phi(x) - F(x)\| < \epsilon.$$

*Proof of Theorem 3.1 (ii).* There exists a compact and permutation-invariant subset $X \subseteq \mathcal{G}_{n,m}$ such that $\mathbb{P}[X] > 1 - \epsilon$ and that for any $(G, H) \in X$, $G$ is connected and has no cycles of length greater than 3. The rest is to apply Theorem A.2 on $X$ and $\mathcal{F} = \mathcal{F}_{1,v}$, for which one needs to verify the four conditions in Theorem A.2.

- *Verification of Condition (i).* By its construction, $\mathcal{F}_{1,v}$ is a subalgebra of $\mathcal{C}(X, \mathbb{R})$. In addition, $\mathbf{1} \in \mathcal{F}_{1,v}$ if the output function $r$ always takes the constant value 1.

- *Verification of Condition (ii).* Notice that $\mathcal{F}_1 \mathbf{1} \subset \mathcal{F}_{1,v}$. If $F(G, H) = F(\hat{G}, \hat{H}), \, \forall \, F \in \mathcal{F}_1$, then Theorem 2.5 implies that for any $F_v \in \mathcal{F}_{1,v}$, one has $F_v(G, H) = \sigma(F_v(\hat{G}, \hat{H}))$ for some permutation $\sigma \in S_n$.

- *Verification of Condition (iii).* Suppose that $F_v(G, H) = F_v(\hat{G}, \hat{H}), \, \forall \, F_v \in \mathcal{F}_{1,v}$. By Theorem 2.5, it holds that $(G, H) \overset{1,v}{\sim} (\hat{G}, \hat{H})$. By Theorem 3.2, we know that $(G, H)$ and $(\hat{G}, \hat{H})$ are isomorphic, i.e., $(G, H) = \sigma * (\hat{G}, \hat{H})$ for some $\sigma \in S_n$, which leads to

$$\Phi_v(G, H) = \Phi_v(\sigma * (\hat{G}, \hat{H})) = \sigma(\Phi_v(\hat{G}, \hat{H})). \tag{A.1}$$

Moreover, it follows from $\sigma * (\hat{G}, \hat{H}) = (G, H) \overset{1,v}{\sim} (\hat{G}, \hat{H})$ and Theorem A.1 that

$$\Phi_v(\hat{G}, \hat{H})_i = \Phi_v(\hat{G}, \hat{H})_{\sigma(i)}, \quad \forall \, i \in \{1, 2, \dots, n\}. \tag{A.2}$$

Then one can conclude $\Phi_v(G, H) = \Phi_v(\hat{G}, \hat{H})$ by combining (A.1) and (A.2).

- *Verification of Condition (iv).* Condition (iv) is a direct corollary of Theorem 2.6 and Theorem A.1.

$\square$

Finally, we present the proof of Theorem 3.2 and Theorem A.1.

*Proof of Theorem 3.2.* Let $\mathcal{A} = (A_1, A_2, \ldots, A_s)$ be an $s$-tuple of subgraphs of a graph $A$, and let $\mathcal{B} = (B_1, B_2, \ldots, B_s)$ be an $s$-tuple of subgraphs of a graph $B$. Let $V(\mathcal{A})$ be the union of the vertices in $A_1, A_2, \ldots, A_s$, and let $V(\mathcal{B})$ be the union of the vertices in $B_1, B_2, \ldots, B_s$. We say that $\mathcal{A}$ and $\mathcal{B}$ are isomorphic if there exists a bijective map of vertices of $V(\mathcal{A})$ to vertices of $V(\mathcal{B})$ such that for any $i \in \{1, 2, \ldots, s\}$,

- all vertices of $A_i$ are mapped to vertices of $B_i$ with the same label/feature and vice versa

- all edges of $A_i$ are mapped to edges of $B_i$ and vice versa.

Consider $(G, H) \overset{1}{\sim} (\hat{G}, \hat{H})$, i.e., $(G, H)$ and $(\hat{G}, \hat{H})$ cannot be distinguished by the 1-hop subgraph WL test. When there are no hash collisions and the colors stabilize, the multisets of final colors of vertices in $G$ and $\hat{G}$ are the same, and any $v_1 \in G$ and $v_2 \in \hat{G}$ with the same color must have isomorphic 1-hop subgraphs.

We abbreviate an induced subgraph of $(G, H)$ or $(\hat{G}, \hat{H})$ as its set of vertices. For any set $S$ of vertices, let $\mathcal{N}(S)$ be the set of all vertices in $S$ or neighboring some vertex of $S$. We prove the following statement by induction: for any $t \in \{1, 2, \ldots, |G|\}$, there exist connected isomorphic subsets $S_1 \subseteq V(G)$ and $S_2 \subseteq V(\hat{G})$ of size $t$, where $V(G)$ and $V(\hat{G})$ are vertex sets of $G$ and $\hat{G}$ respectively, such that $(S_1, \mathcal{N}(S_1))$ and $(S_2, \mathcal{N}(S_2))$ are isomorphic. For the base case, choose any two vertices in $G$ and $\hat{G}$ with the same color. For the inductive step, suppose that $S_1$ and $S_2$ are sets of size $t < |V(G)|$, and we want to find two sets $S_1'$ and $S_2'$ with size $t + 1$ that satisfy the inductive statement. Let $v_1$ be a vertex not in $S_1$ adjacent to a vertex in $S_1$, and let $v_2$ be the image of $v_1$ under the isomorphism $f : (S_1, \mathcal{N}(S_1)) \to (S_2, \mathcal{N}(S_2))$, i.e., $v_2 = f(v_1)$. Let $\mathcal{N}(v_1)$ and $\mathcal{N}(v_2)$ be the sets of vertices with distance at most 1 from $v_1$ and $v_2$, respectively. Then $\mathcal{N}(v_1)$ and $\mathcal{N}(v_2)$ are isomorphic since $v_1$ and $v_2$ are of the same color. We aim to show that $f$ can be extended to an isomorphism from $(S_1 \cup \{v_1\}, \mathcal{N}(S_1) \cup \mathcal{N}(v_1))$ to $(S_2 \cup \{v_2\}, \mathcal{N}(S_2) \cup \mathcal{N}(v_2))$.

Consider $T_1 = \mathcal{N}(v_1) \backslash \mathcal{N}(S_1)$ and $T_2 = \mathcal{N}(v_2) \backslash \mathcal{N}(S_2)$. We claim that any vertex $u_1$ of $T_1$ cannot be connected to a vertex of $\mathcal{N}(S_1)$ other than $v_1$. If $u_1$ is connected to some vertex $u_2 \neq v_1$ in $\mathcal{N}(S_1)$, then both $v_1$ and $u_2$ must have some neighbor in $S_1$: call these $u_3$ and $u_4$. If $u_3 = u_4$, then we have the cycle $u_1 \to v_1 \to u_3 \to u_2 \to u_1$. If $u_3 \neq u_4$, then there must be a path through edges of $S_1$ from $u_3$ to $u_4$, so we create a cycle containing $u_1 \to v_1 \to u_3 \to \cdots \to u_4 \to u_2 \to u_1$. Both of these cycles have a length greater than 3, which is a contradiction. Thus, $u_1$ is not connected to any vertex of $\mathcal{N}(S_1)$ other than $v_1$. Similarly, any vertex of $T_2$ is not connected to a vertex of $\mathcal{N}(S_2)$ other than $v_2$.

Another observation is that in the induced subgraph of $T_1$ (or $T_2$), the degree of each vertex is at most 1. In particular, if $u_1 \in T_1$ is connected to $u_2, u_3 \in T_1$ with $u_2 \neq u_3$, then there is a cycle $u_2 \to u_1 \to u_3 \to v_1 \to u_2$ of length 4, which is a contradiction.

Notice that $f : (S_1, \mathcal{N}(S_1)) \to (S_2, \mathcal{N}(S_2))$ is an isomorphism and that $\mathcal{N}(S_1) \cap \mathcal{N}(v_1)$ is isomorphic to $\mathcal{N}(S_2) \cap \mathcal{N}(v_2)$. It can be seen that the multisets of vertex colors in $T_1$ and $T_2$ are the same. Additionally, edges connecting vertices in $T_1$ can be paired with edges connecting vertices in $T_2$, so that the paired edges have the same multiset of end vertex features. Since no such edges share a common end vertex, guaranteed by the above observation, one can extend $f$ to an isomorphism from $(S_1 \cup \{v_1\}, \mathcal{N}(S_1) \cup \mathcal{N}(v_1))$ to $(S_2 \cup \{v_2\}, \mathcal{N}(S_2) \cup \mathcal{N}(v_2))$. Thus, we have proven the inductive step and the proof is completed. $\square$

*Proof of Corollary A.1.* By the proof of Theorem 3.2, there exists a permutation $\sigma \in S_n$ such that $\sigma(i) = i'$ and $\sigma * (G, H) = (G, H)$. Then the result holds immediately. $\square$

# B    PROOFS FOR SECTION 3.2

*Proof of Theorem 3.4.* Based on Theorem 3.5, the proof of Theorem 3.4 follows the same lines as the proof of Theorem 3.1. □

Next, we present the proof of Theorem 3.5. Let $S$ be a subset of vertices of a graph. Define $\mathcal{N}_k(S)$ as the set of all vertices with distance at most $k$ from any vertex in $S$ and $\mathcal{N}_k(v)$ as the set of all vertices with distance at most $k$ from $v$. If $S$ is nonempty, define $d(v, S)$ as the minimum distance from $v$ to any vertex in $S$.

To prove our Theorem 3.5, we need a lemma, which rules out the existence of undetected edges when we do our induction.

**Lemma B.1.** *Let $k \geq 2$ and let $S$ be a connected subset of vertices of a connected graph $G$ with no cycles of length greater than $2k + 1$. Let $u_1$ be a vertex not in $S$ adjacent to a vertex in $S$. Then, no vertex in $T = \mathcal{N}_k(u_1) \setminus \mathcal{N}_k(S)$ can be connected to a vertex in $\mathcal{N}_k(S) \setminus \mathcal{N}_k(u_1)$.*

*Proof.* Assume for the sake of contradiction that there exists a vertex $u_{k+1} \in T$ connected to $v_k \in \mathcal{N}_k(S) \setminus \mathcal{N}_k(u_1)$. Notice that $d(v_k, S) \leq k$ because $v_k \in \mathcal{N}_k(S)$. If $d(v_k, S) < k$, then $d(u_{k+1}, S) \leq k$, which contradicts $u_{k+1} \notin \mathcal{N}_k(S)$. Thus, $d(v_k, S) = k$.

Therefore, there must exist vertices $u_2, u_3, \ldots, u_k$ and $v_0, v_1, \ldots, v_{k-1}$ such that $u_i$ and $u_{i+1}$ are connected for $i \in \{1, 2, \ldots, k\}$, $v_i$ and $v_{i+1}$ are connected for $i \in \{0, 1, \ldots, k-1\}$, and $v_0 \in S$.

We claim that $u_1, u_2, \ldots, u_{k+1}, v_0, \ldots, v_k$ are pairwise distinct. For any two connected vertices $a$ and $b$, notice that $|d(a, S) - d(b, S)| \leq 1$ because any path of length $s$ from $a$ to a vertex of $S$ can be extended to a path of length $s + 1$ from $b$ to a vertex of $S$ and vice versa. Since $d(u_1, S) = 1$, $d(u_{k+1}, S) = k + 1$, $d(v_0, S) = 0$, and $d(v_k, S) = k$, we must have $d(u_i, S) = i$ and $d(v_i, S) = i$ for all valid $i$. Thus, the only possible pairs of vertices of $u_1, u_2, \ldots, u_{k+1}, v_0, \ldots, v_k$ that can be equal are $(u_i, v_i)$ for $i \in \{1, 2, \ldots, k\}$. Assume for the sake of contradiction that $u_i = v_i$ for some $i$. Then, there exists a path $u_1 \to u_2 \to \cdots \to u_i \to v_{i+1} \to \cdots \to v_k$ of length $k - 1$ from $u_1$ to $v_k$, contradicting the fact that $v_k \notin \mathcal{N}_k(u_1)$. Thus, the vertices $u_1, u_2, \ldots, u_{k+1}, v_0, \ldots, v_k$ are pairwise distinct.

Since $S$ is connected, there exists a path with edges in $S$ from $v_0$ to a vertex in $S$ adjacent to $u_1$. We can combine this path with $u_1 \to u_2 \to \cdots \to u_{k+1} \to v_k \to v_{k-1} \to \cdots \to v_0$ to create a cycle containing vertices $u_1, u_2, \ldots, u_{k+1}, v_0, \ldots, v_k$. This cycle contains at least $2k + 2$ vertices, a contradiction. □

*Proof of Theorem 3.5.* We work with the same notation and setting as in the proof of Theorem 3.2. Consider $(G, H) \overset{k}{\sim} (\hat{G}, \hat{H})$, i.e., $(G, H)$ and $(\hat{G}, \hat{H})$ cannot be distinguished by the $k$-hop subgraph WL test. When there are no hash collisions and the colors stabilize, the multisets of final colors of vertices in $G$ and $\hat{G}$ are the same, and any $v_1 \in G$ and $v_2 \in \hat{G}$ with the same color must have isomorphic $k$-hop subgraphs rooted at them.

We prove the following statement by induction: for any $t \in \{1, 2, \ldots, |G|\}$, there exist connected isomorphic subsets $S_1 \subseteq V(G)$ and $S_2 \subseteq V(\hat{G})$ of size $t$ such that $(S_1, \mathcal{N}_k(S_1))$ and $(S_2, \mathcal{N}_k(S_2))$ are isomorphic. For the base case, choose any two vertices in $G$ and $\hat{G}$ with the same color. For the inductive step, suppose that $S_1$ and $S_2$ are valid sets of size $t < |V(G)|$, and we want to find two sets $S_1'$ and $S_2'$ with size $t + 1$ satisfying the inductive statement. Let $v_1$ be a vertex not in $S_1$ adjacent to a vertex in $S_1$, and let $v_2$ be the image of $v_1$ under an isomorphism $f$ from $(S_1, \mathcal{N}_k(S_1))$ to $(S_2, \mathcal{N}_k(S_2))$. Then $\mathcal{N}_k(v_1)$ and $\mathcal{N}_k(v_2)$ are isomorphic since $v_1$ and $v_2$ are of the same color, and $f$ takes $\mathcal{N}_k(S_1) \cap \mathcal{N}_k(v_1)$ to $\mathcal{N}_k(S_2) \cap \mathcal{N}_k(v_2)$.

The $k$-separability assumption guarantees that vertices in $\mathcal{N}_k(v_i) \setminus \mathcal{N}_k(S_i)$ have distinct colors for $i = 1, 2$. Thus, there is a unique way to extend $f$ as a map from $\mathcal{N}_k(S_1) \cup \mathcal{N}_k(v_1)$ to $\mathcal{N}_k(S_2) \cup \mathcal{N}_k(v_2)$, which keeps that a vertex has the same color as its image. We then verify that this extension is still an isomorphism. Consider any $u_1 \in \mathcal{N}_k(v_1) \setminus \mathcal{N}_k(S_1)$ and any $w_1 \in \mathcal{N}_k(v_1)$ with $u_1 \neq w_1$. Denote $u_2 = f(u_1)$ and $w_2 = f(w_1)$. Then $d(u_1, v_1) = d(u_2, v_2) = k$. We claim that $u_1$ and $w_1$ are connected if and only if $u_2$ and $w_2$ are connected.

- Case 1: $d(w_1, v_1) = d(w_2, v_2) = k$. By the $k$-separability assumption and the isomorphism between $\mathcal{N}_k(v_1)$ and $\mathcal{N}_k(v_2)$, we immediately have that $u_1$ and $w_1$ are connected if and only if $u_2$ and $w_2$ are connected.

- Case 2: $d(w_1, v_1) = d(w_2, v_2) = k - 1$. Note that the multisets of vertex colors of the direct neighbors of $w_1$ and $w_2$ are the same, which combined with the $k$-separability assumption that $u_1$ and $w_1$ are connected if and only if $u_2$ and $w_2$ are connected.

- Case 3: $d(w_1, v_1) = d(w_2, v_2) \leq k - 2$. Then $u_i$ and $w_i$ are not connected for $i = 1, 2$.

By Theorem B.1 and the above arguments, we conclude that $f$ is an isomorphism from $(S_1 \cup \{v_1\}, \mathcal{N}_k(S_1) \cup \mathcal{N}_k(v_1))$ to $(S_2 \cup \{v_2\}, \mathcal{N}_k(S_2) \cup \mathcal{N}_k(v_2))$. This completes the inductive step. $\quad\square$

## C  PRELIMINARY RESULTS ON $k$-HOP SPD-GNNS

In this section, we present some preliminary results on the expressive power of $k$-hop subgraph GNNs where the information on the $k$-hop subgraphs is processed and aggregated in a specific way, as in (4.1). These results can be viewed as extensions of Section 3.

In particular, we name such GNN architecture as $k$-hop SPD-GNN that is defined as follows for $(G, H) \in \mathcal{G}_{n,m}$ and $k \geq 1$.

- The embedding layer maps each vertex feature $h_i \in \mathbb{R}^m$ as an embedding vector

$$h_i^{(0)} = f^{(0)}(h_i),$$

  where $f^{(0)}$ is learnable.

- For $l = 1, 2, \ldots, L$, the information aggregation layer computes $h_i^{(l)}$ according to (4.1), where $d(\cdot, \cdot)$ is the shortest-path distance (SPD), for $i = 1, 2, \ldots, n$.

- There are two types of outputs. The graph-level output computes a real number for the whole graph, namely

$$y = r\left(\text{AGGR}\left(\left\{\left\{h_i^{(L)} : i \in \{1, 2, \ldots, n\}\right\}\right\}\right)\right),$$

  where $r$ is learnable. The vertex-level output assigns a real number for each vertex:

$$y_i = r(h_i^{(L)}), \quad i = 1, 2, \ldots, n.$$

**Definition C.1** (Spaces of $k$-hop SPD-GNNs). *We use $\mathcal{F}_k^{\text{spd}}$ to denote the collection of all $k$-hop SPD-GNNs with graph-level output, and use $\mathcal{F}_{k,v}^{\text{spd}}$ to denote the collection of all $k$-hop SPD-GNNs with vertex-level output.*

It is clear that the implementation of $k$-hop SPD-GNNs is cheaper than general $k$-hop subgraph GNNs as some topology information about the subgraph is dropped. The trade-off is that $k$-hop SPD-GNNs have a bit weaker expressive/separation power. To rigorously introduce our next theorem on the expressive power of $k$-hop SPD-GNNs, we require the associated $k$-hop SPD-WL test implementing the color refinement as follows:

$$C^{(l)}(v_i) = \text{HASH}\left(C^{(l-1)}(v_i), \left\{\left\{\left(C^{(l-1)}(v_j), d(v_i, v_j)\right) : v_j \in \mathcal{N}_k(v_i)\right\}\right\}\right).$$

**Definition C.2** ($k$-strongly separable graph). *A graph $(G, H) \in \mathcal{G}_{n,m}$ is said to be $k$-strongly separable if the following condition holds when the $k$-hop SPD-WL test terminates with stabilized colors and without hash collisions: For any two vertices $v_1, v_2$ with $d(v_1, v_2) \leq 2k$, the colors of $v_1$ and $v_2$ are different.*

Our main result in this section is that $k$-hop SPD-GNNs can universally approximate any permutation-invariant/equivariant continuous functions on $k$-strongly separable graphs with no cycles of length greater than $2k - 1$.

**Theorem C.3.** *Let $k \geq 2$ and let $\mathbb{P}$ be a Borel probability measure on $\mathcal{G}_{n,m}$. Suppose that $\mathbb{P}$-almost surely, $(G, H)$ is $k$-strongly separable and $G$ is connected with no cycles of length greater than $2k - 1$. Then, the following hold.*

(i) *For any $\epsilon, \delta > 0$ and any permutation-invariant continuous function $\Phi : \mathcal{G}_{n,m} \to \mathbb{R}$, there exists $F \in \mathcal{F}_k^{\mathrm{spd}}$ such that*

$$\mathbb{P}\left[|F(G, H) - \Phi(G, H)| > \delta\right] < \epsilon.$$

(ii) *For any $\epsilon, \delta > 0$ and any permutation-equivariant continuous function $\Phi_v : \mathcal{G}_{n,m} \to \mathbb{R}^n$, there exists $F_v \in \mathcal{F}_{k,v}^{\mathrm{spd}}$ such that*

$$\mathbb{P}\left[\|F_v(G, H) - \Phi_v(G, H)\| > \delta\right] < \epsilon.$$

*Proof of Theorem C.3.* Based on Theorem C.4 below, stating that the $k$-hop SPD-WL test can distinguish all non-isomorphic $k$-strongly separable graphs, and analogues to Theorem 2.5 and Theorem 2.6, one can prove Theorem C.3 following the same lines in the proof of Theorem 3.1. □

**Theorem C.4.** *Let $k \geq 2$ and consider $k$-strongly separable graphs $(G, H), (\hat{G}, \hat{H}) \in \mathcal{G}_{n,m}$. Suppose that $G$ and $\hat{G}$ are both connected and have no cycles of length greater than $2k - 1$. If $(G, H)$ and $(\hat{G}, \hat{H})$ are indistinguishable by the $k$-hop SPD-WL test, then they must be isomorphic.*

*Proof of Theorem C.4.* The main idea is that under the assumption of strong $k$-separability, the $k$-hop SPD-WL test has stronger separation power than $(k - 1)$-hop SPD-WL test, and hence we can apply the results in Theorem 3.2 and Theorem 3.5.

In particular, let us consider the final stabilized colors on $(G, H)$ and $(\hat{G}, \hat{H})$ generated by $k$-hop SPD-WL test ($k \geq 2$) without hash collisions. For any vertex $v$, notice that any pair of vertices in $\mathcal{N}_k(v)$ have a distance at most $2k$ from each other, so they are of different colors. Suppose $u_1$ and $u_2$ are vertices in $\mathcal{N}_{k-1}(v)$. If the color of $u_1$ implies it has a neighbor with the same color as $u_2$, then this neighbor must be $u_2$, as the only neighbors of $u_1$ are in $\mathcal{N}_k(v)$ and all vertices in $\mathcal{N}_k(v)$ are of different colors. Otherwise, $u_1$ and $u_2$ cannot be connected by edges. Thus, for any $u_1$ and $u_2$ in $\mathcal{N}_{k-1}(v)$, we can uniquely determine whether there is an edge between $u_1$ and $u_2$ by their colors. This implies that two vertices of the same color must have isomorphic $(k - 1)$-hop subgraphs, i.e., implementing $(k - 1)$-hop subgraph WL test does not lead to a strict color refinement. Then the result is a direct corollary of Theorem 3.2 and Theorem 3.5. □

It is worth noting that when $k = 1$, Theorem C.4 holds even without the 1-strong separability assumption. This is because that 1-hop SPD-WL test coincides with the classic WL test and it is proved in Bamberger (2022) that the classic WL test can distinguish any non-isomorphic trees. In this sense, Theorem C.4 can be regarded as an extension of Bamberger (2022). Another related work is Feng et al. (2022) that proves that the separation power of the $k$-hop SPD-WL test is upper bounded by that of the third-order WL test.

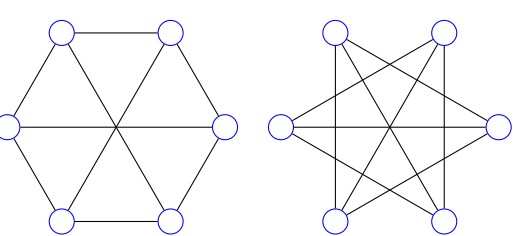

Figure 6: The $k$-strong separability assumption is necessary in Theorem C.3

We also comment that Theorem C.4 does not hold true if the $k$-strong separability condition is removed. Consider the two graphs in Figure 6, in which all vertices have the same initial feature. Each vertex has three neighbors of distance 1, two neighbors of distance 2, and no neighbors of higher distance, so all vertices would have the same color in any $k$-hop SPD-WL test for any positive integer $k$. Thus, these two non-isomorphic graphs cannot be distinguished by $k$-hop SPD-WL test for any $k \geq 1$, which does not contradicts Theorem C.4 as these two graphs are clearly not $k$-strongly separable. This example also illustrates that it is impossible to prove the conclusion of Theorem C.4 without any condition.

