# OpenReview forum: "Expressive Power of Subgraph Graph Neural Networks for Graphs with Bounded Cycles"
_ICLR.cc/2026/Conference — ICLR 2026 Conference Withdrawn Submission_

### Official Review · Reviewer_zp9J · 2025-10-27

**Soundness:** 3
**Presentation:** 4
**Contribution:** 2
**Rating:** 4
**Confidence:** 4

**Summary:**

This paper analyzes the expressive power of $k$-hop subgraph GNNs. It shows that there exist $k$-hop subgraph GNN instances capable of distinguishing any pair of non-isomorphic $k$-separable graphs whose cycles have length at most $2k+1$. Consequently, these models can approximate any continuous permutation invariant or permutation equivariant function over such graphs. Empirical results on the ZINC molecular dataset indicate that performance improves significantly for $k>1$ and further benefits from using resistance distance instead of shortest path distance.

**Strengths:**

- While the expressive power of k-hop subgraph GNNs has been explored in prior work, to the best of my knowledge the results presented in this paper are novel. This paper focuses on $k$-separable graphs with bounded cycle length which have not been explored in previous studies.

- The empirical results are consistent with the theory. The 1-hop Graphormer is significantly outperformed by models that consider larger subgraphs, and replacing the shortest path distance with resistance distance further improves performance.

- The paper is well-written and easy to read.

**Weaknesses:**

- More focus should be placed on the definition of the $k$-separable graphs. Since Theorems 3.4 and 3.5 assume $k$-separable graphs, the paper should provide additional details on which graphs belong to this class, along with illustrative examples of both $k$-separable and non-$k$-separable graphs.

- The theoretical results are not validated by numerical experiments. I would suggest the authors conduct further experiments to validate that the model more expressive than GNNs whose power is upper-bounded by 1-WL. A dataset needs to be constructed that contains non-isomorphic $k$-separable graphs that contain cycles of length smaller than $2k+1$ and which cannot be distinguished by standard GNNs.

- The motivation of the paper could be strengthened. It is not clear why the proposed results are more significant than previous expressivity results on subgraph GNNs. Do common benchmark datasets consist of $k$-separable graphs with bounded cycle lengths? That would make the analysis useful.

- The organization of the paper could be significantly improved. The first 5 pages of the paper mainly cover well-known material, and could be shortened to make room for a more thorough empirical validation.

- The discussion of related work is incomplete. For instance, the paper does not cite the first GNN that incorporates k-hop subgraph structures [1] and which preceded [2] and [3]. Other subgraph GNN models are also not discusses such as the one presented in [4] which relates subgraph GNNs to the $k$-WL hierarchy.

[1] Nikolentzos, G., Dasoulas, G., & Vazirgiannis, M. K-hop graph neural networks. Neural Networks, Vol. 130, pp. 195-205, 2020.\
[2] Zhang, M., & Li, P. Nested graph neural networks. Advances in Neural Information Processing Systems, pp. 15734-15747, 2021.\
[3] Feng, J., Chen, Y., Li, F., Sarkar, A., & Zhang, M. How powerful are k-hop message passing graph neural networks. Advances in Neural Information Processing Systems, pp. 4776-4790, 2022.\
[4] Qian, C., Rattan, G., Geerts, F., Niepert, M., & Morris, C. Ordered subgraph aggregation networks. Advances in Neural Information Processing Systems, pp. 21030-21045, 2022.

**Questions:**

- Why is the Graphormer backbone chosen instead of a standard GNN model for the empirical evaluation?

---

### Official Review · Reviewer_qsHS · 2025-10-28

**Soundness:** 3
**Presentation:** 2
**Contribution:** 2
**Rating:** 4
**Confidence:** 4

**Summary:**

This paper analyzes the expressive power of $k$-hop subgraph GNNs for graphs with bounded cycle lengths, where  $k$-hop subgraph GNNs aggregate information from subgraphs within distance $k$ from a vertex. The authors prove that these GNNs can distinguish all non-isomorphic connected graphs without cycles longer than $2k+1$ and can approximate any continuous permutation-invariant or equivariant function on such graphs. Empirically, the authors validate their findings on the ZINC molecular graph dataset using modified Graphormer, showing that performance improves significantly up to $k=4$ and saturates thereafter, aligning with the dataset's cycle length distribution.

**Strengths:**

1. This paper provides a theoretical analysis of the expressive power of $k$-hop subgraph GNNs for graphs with bounded cycles, extending prior work on MP-GNNs and WL tests.
2.  Experiments on the ZINC dataset support the theoretical predictions, where performance improves as $k$ increases, and the use of both shortest-path and resistance distance variants shows the robustness of the $k$-hop framework.
3.  The paper is well-organized, with clear definitions. The authors also dicuss the limitations of the current work.

**Weaknesses:**

1. First of all, this paper needs a better motivation. Otherwise, the results are quite niche. Subgraph GNNs are well-studied, and there are a ton of recent papers on it. The authors need to explain how this work build/extends on previous findings, and why the results in this paper are significant. Also, I suggest a related work section to be added, providing more discussions on existing literatures.
2. Experiments are conducted only on the ZINC dataset, which is known to have small graphs and bounded cycles. More diverse benchmarks (e.g., social networks with longer cycles) would strengthen the claims. Also, ablation on the impact of different subgraph encoding schemes will further validate the results.
3. Computational cost is not discussed, which relates to the practical importance of the proposed framework.

**Questions:**

1. This paper evaluates $k$-hop graphormer particularly, how about the performance for other $k$-hop GNNs?
2. What is the computational complexity of $k$-hop subgraph GNNs, especially as $k$ increases?
3. How does the model perform on datasets with larger cycles or more complex structures?
4. How does the $k$-hop subgraph gnn compare with other higher-order GNNs (e.g., Hypergraph, Simplicial Complex) in terms of both expressive power and computational cost?

---

### Official Review · Reviewer_xww7 · 2025-10-31

**Soundness:** 4
**Presentation:** 4
**Contribution:** 4
**Rating:** 8
**Confidence:** 4

**Summary:**

In the submitted manuscript, the authors analyse the expressive power of k-hop subgraph GNNs and, under certain assumptions, manage to prove that these GNNs can separate graphs without cycles of length greater than 2k+1. These strong theoretical results are supported by empirical results on the ZINC dataset.

**Strengths:**

- I very much appreciated the clarity of your writing. Having a running example in Sections 1 and 2 is great.
- Clearly, a substantial amount of theoretical progress is presented in this paper.
- While the need for more and more expressive models is recently being questioned in our literature, I still see great value in studies such as the submitted, that complete our understanding of the expressive power of the different GNN variants.

**Weaknesses:**

- As for many papers that mostly make theoretical contributions, one could ask for experiments on more datasets. But since the theoretical side of this work seems rather strong to me, I don't see a great need for that in this paper.

- Some of your definitions/assumptions and theoretical results could be supplemented by more discussion (see my questions for the particular instances I am talking about).

**Questions:**

1] Your theoretical results mostly require the graph to be connected. Yet this assumption is not discussed in your paper. Could you comment on why this is required? And may it be useful to include a brief remark on this in the paper?

2] Definition 3.3 is currently presented as a purely technical assumption without any discussion. Could you explain why this assumption is required for your proofs and give some intuition on how restrictive this assumption is? Are many graphs in the dataset that you use, the ZINC dataset, k-separable?

3] I think it would be nice if the authors could include the proof of Theorem 2.5 in the appendix. Even if it is straightforward, it seems to me that writing it out would solidify the basis that your work stands on.

4] The result of Corollary 2.6 is not discussed or interpreted at all. It may be nice to add one or two sentences contextualising the result and interpreting it.

5] In Figure 5 the MAE for k=1 is not visible in the plot. Yet you discuss this value in particular in the text. Would it be possible to modify the plot so that this value is visible?

6] Minor Comments:

6.1] When discussing the WL test in Line 82 you say "It deems two graphs isomorphic if their final color multisets match." I think this should be slightly adapted, since the test is inconclusive if the multisets match and rejects isomorphism if the multisets mismatch.

6.2] I quite like your example in Figure 3. I think this counterexample could potentially be made more visible by formalising your statement in Lines 339-347 as a Proposition.

6.3] In your conclusion you say that your result holds "unconditionally", technically speaking this is false since you assume the graph to connected. I think it may be nice to add this nuance here.

---

### Official Review · Reviewer_wPfy · 2025-10-31

**Soundness:** 2
**Presentation:** 3
**Contribution:** 2
**Rating:** 2
**Confidence:** 4

**Summary:**

The article investigates the expressive power of k-hop subgraph GNNs for graphs with bounded cycle length. The main result states that k-hop subgraph GNNs can uniquely (up too isomorphism) identify a graph if it has no cycles greater than 2k+1 and is k-separable, a graph property that that is fulfilled if a certain kind of pairwise node label inequality holds. In an experimental study, the authors show that applying their theory to extend the Graphormer model leads to improvements for certain values of k on the ZINC dataset.

**Strengths:**

1. The article is easy to follow and overall of good technical quality.
2. The authors provide formal proves with proper theoretical guarantees extending previous results.
3. Theorems 3.2/3.5 imply a nice canonicalisation for the class of k-separable bounded cycle length graphs.

**Weaknesses:**

1. The overall contribution is limited. While the article provides a nice characterisation of a specific graph class, the results provide only modest additional insights into the expressive power of (subgraph) GNNs.
2. The notion of k-separability plus bounded cycle length is quite a strong assumption such that the practical relevance of the results is not immediate.
3. I believe the graphs of figure 3 are distinguishable even with 1-hop subgraph GNNs. If so, the given example is rather not an ideal showcase.
4. The expressivity aspect is poorly evaluated in practice. I would advise the authors to at least analyse how often the theoretical conditions of 3.2/3.5 are met.
5. The experimental evaluation is largely based on only a single graph benchmark dataset.
6. It is questionable whether the expressivity results of this paper are of relevance to ZINC, since even ordinary WL is generally sufficient to distinguish most molecular graphs.
7. The results of Table 1 show only marginal improvements.

**Questions:**

1. How common are graphs in practice that are k-separable and contain no cycles of length larger 2k+1 but cannot be distinguished by ordinary WL nor (k-1)-hop subgraph GNNs?
2. Do you have insights into the fraction of graphs in standard benchmark datasets that can be uniquely identified using theorems 3.2/3.5 (for different choices of k)?
3. Can you show that the performance improvements actually come from increased expressivity?
4. Do you have insights into when the k-separability is naturally satisfied? Similarly, are there  natural graph classes which can be uniquely identified up to isomorphism using Theorem 3.2/3.5?
5. Do you have an intuition why k-separability is not necessary for k=1 but for k>=2?

---

### Note · Authors · 2025-11-26

**Comment:**

We sincerely appreciate the reviewers’ constructive feedback. After careful consideration, we have decided to withdraw this paper and will work to address the comments in future revisions.

**Withdrawal Confirmation:**

I have read and agree with the venue's withdrawal policy on behalf of myself and my co-authors.